# Confinement-induced accumulation and de-mixing of microscopic active-passive mixtures

Stephen Williams [1], Raphaël Jeanneret [2], Idan Tuval [3,4] & Marco Polin[1,3,4] ✉

Understanding the out-of-equilibrium properties of noisy microscale systems and the extent to which they can be modulated externally, is a crucial scientific and technological challenge. It holds the promise to unlock disruptive new technologies ranging from targeted delivery of chemicals within the body to directed assembly of new materials. Here we focus on how active matter can be harnessed to transport passive microscopic systems in a statistically predictable way. Using a minimal active-passive system of weakly Brownian particles and swimming microalgae, we show that spatial confinement leads to a complex non-monotonic steady-state distribution of colloids, with a pronounced peak at the boundary. The particles' emergent active dynamics is well captured by a space-dependent Poisson process resulting from the space-dependent motion of the algae. Based on our findings, we then realise experimentally the de-mixing of the active-passive suspension, opening the way for manipulating colloidal objects via controlled activity fields.

Evolution has enabled living systems to achieve an exquisite control of matter at the microscopic level. From the precise positioning of chromosomes along the mitotic spindle[1] to the many types of embryonic gastrulation[2] cells harness their internal and external motility to reach a predictable order despite the stochasticity intrinsic to the microscopic realm[3]. Understanding how order emerges in these active systems is a fundamental scientific challenge with the potential to bring disruptive technologies for the macroscopic control of microscopic structures. Here we address this problem within a minimal active-passive experimental model system.

From a physical perspective, living systems fall under the general category of active matter[4], characterised by emergent phenomena including flocking[5–7], active turbulence[8–11] and motility-induced phase separation[12,13]. As our understanding of single-species active systems progresses, attention has started to veer towards more complex cases where components with different levels of activity interact. Indeed this is often the case for biological active matter. Intracellular activity, for example, can be used for spatial organisation of passive intracellular organelles[14–16]; and clustering induced by motility differentials helps

bacterial swarms expand despite antibiotic exposure[17]. In order to study the emergent properties of these complex systems, an important and phenomenologically rich minimal model is one that mixes active and inert agents. Active impurities have been used to alter the dynamics of grain boundaries in colloidal crystals[18,19] and favour the formation of metastable clusters in semi-dilute suspensions[20]. Sufficiently large concentrations of both active and passive species often reveal a rich spectrum of phases depending on the interactions between constituents[21–29].

Active baths, where individual passive inclusions are dispersed within an active suspension, are particularly appealing. Their conceptual simplicity makes them a natural starting point to develop a statistical theory of active transport[30–33] with potential applications to micro-cargo delivery, micro-actuation[34–36] and nutrient transport[37,38]. In general, passive particles in homogeneous and isotropic active baths display enhanced diffusion due to a continuous energy transfer from the active component via direct collisions and hydrodynamic interactions[39–47]. Designing larger objects with asymmetric shapes can then turn active diffusion into noisy active translation or rotation[30,48,49].

[1]Department of Physics, University of Warwick, Coventry CV4 7AL, United Kingdom. [2]Laboratoire de Physique de l'Ecole Normale Supérieure, ENS, Université PSL, CNRS, Sorbonne Université, Université de Paris, F-75005 Paris, France. [3]Departament de Física, Universitat de les Illes Balears, 07071 Palma de Mallorca, Spain. [4]Instituto Mediterráneo de Estudios Avanzados, IMEDEA, Miquel Marques 21, 07190 Esporles, Spain. ✉e-mail: mpolin@imedea.uib-csic.es

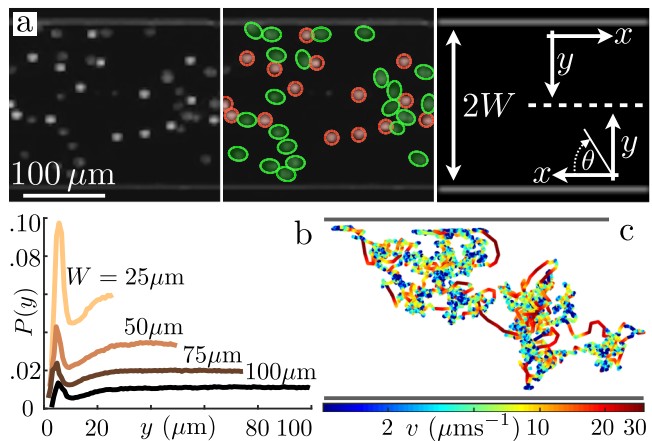

**Fig. 1 | Experimental setup and colloidal behaviour. a** Dark-field image of cells and beads within a straight microfluidic channel (here width $2W = 200\,\mu m$). The two species are highlighted with, respectively, green ellipses and red circles. The schematic illustrates the coordinate system used throughout. **b** Steady-state colloidal probability distribution functions across the channel (edge to midpoint) for different $W$ values. Boundary accumulation is followed by a depleted adjacent region which gives way to a uniform distribution within the channel. **c** Example of a typical colloidal trajectory inside a $100\,\mu m$-width channel. Colours represent the mean frame-to-frame speed. The dynamics is composed of slow diffusive-like motion (blue sections) interspersed with fast straighter jumps (red sections).

However the strategies for the predictable patterning and transport of passive cargo are still very limited. This is in contrast with the exquisite level of external control possible on the motility of biological and synthetic active particles themselves[50–55], such as photokinetic bacteria whose swim speed depends on the local light intensity; property that can be harnessed to pattern surfaces[53,54].

Here we show that a steady gradient of activity within active-passive suspensions can be harnessed to control the fate of generic passive particles. Quasy-2D microfluidic channels are filled with a binary suspension of polystyrene colloids and unicellular biflagellate microalgae *Chlamydomonas reinhardtii* (see Supplementary Movie 1). Confinement induces a spatially inhomogeneous and anisotropic distribution of microswimmers as a result of wall scattering[56–59], which translates into a space-dependent active noise for the colloids. We show that this produces complex non-monotonic colloidal distributions with accumulation at the boundaries and then develop an effective microscopic jump-diffusion model for the colloidal dynamics. The latter shows excellent analytical and numerical agreement with the experimental results. Finally we demonstrate how confinement with aptly designed microfluidic chips can fuel the de-mixing of active-passive suspensions, opening the way for manipulating passive colloidal objects via controlled activity fields.

## Results

### Colloid concentration is non-homogeneous under confinement

A full description of the experimental procedures (culturing, video microscopy and data analysis) can be found in the Methods section and Supplementary Materials (data and codes also available on Zenodo[60]). Figure 1a shows a detail of the main section of the first type of microfluidic setup used. These devices, between $14\,\mu m$ and $20\,\mu m$ thick, are composed of two reservoir chambers connected by long and straight channels of constant width $2W$ ranging from 50 to $200\,\mu m$. These are filled with a dilute mixture of *Chlamydomonas reinhardtii* (CR; strain CC-125, radius $R \sim 4$–$5\,\mu m$) and weakly Brownian colloids (radius $a = 5 \pm 0.5\,\mu m$) at surface fractions $\phi_{CR} \approx \phi_{col} \approx 2$–3% (bulk concentrations ~2–$3 \times 10^7$ particles/mL). The design ensures a steady concentration along the connecting channels for both species (see Supplementary Movie 1). Figure 1a shows the coordinate system

employed, which has been symmetrized with respect to the channel axis.

As typical of self-propelled particles, the algae tend to accumulate at the boundaries due to their interactions with the side walls[56–58]. This appears as a significant peak in their time-averaged concentration profile at a position $y_{CR} \approx 15\,\mu m$ (Supplementary Fig. 1), roughly equivalent to the sum of the cell radius and the flagellar length. Unexpectedly, we find that also the colloids explore the available space in an inhomogeneous way. Their steady-state distribution (Fig. 1b) shows a clear peak at about one particle radius ($y_{col} = 5.9 \pm 1.5\,\mu m$), followed by a depleted region between $10\,\mu m$ and $20\,\mu m$ from the wall, which roughly corresponds to the peak in algal density, before plateauing to a uniform concentration further inside the channels. This effect is also observed within circular chambers (Supplementary Fig. 2) suggesting it is a robust feature of the system. These distributions are in stark contrast with the equilibrium case (i.e., without microswimmers) for which the colloids are expected to be uniformly distributed despite spatial variations in colloidal diffusivity due to hydrodynamics[61] (Supplementary Fig. 3).

As a first step to gain insight into the colloids' experimental distributions, we characterise their dynamics within our active bath. Figure 1c shows a typical colloidal trajectory (561 s) colour-coded for the average frame-to-frame speed $v$. The dynamics can be understood as a combination of periods of slow diffusive-like displacements ($v \sim 5\,\mu m/s$;) and fast longer jumps ($v \sim 30\,\mu m/s$; see Supplementary Section 1). The latter is reminiscent of hydrodynamic entrainment events reported for micron-sized colloids[47,62], although for these larger particles the prominent role appears to be played by direct flagellar interactions (see Supplementary Movie 2). Irrespective of their specific origin, jump events dominate the active transport of colloids within the system. Following[47,63], we use time-correlation of displacements along individual trajectories and an estimate for the expected magnitude of frame-to-frame diffusive displacements to identify active colloidal jumps and extract their statistical properties as a function of starting position $y$ (we assume translational invariance along the channels' axis). Figure 2a shows the space-dependent distributions of waiting times between successive jumps, with colours representing the distance from the boundary. Regardless of the position, all the distributions are exponential above ~3 s but deviate from it at shorter times. This deviation from a simple Poisson process is due to the large colloidal size influencing the motility of nearby algae as reported also in the case of bacteria[64]. Here, this can lead to a rapid succession of interactions with the same cell (see Supplementary Movie 3). Nevertheless, for our purposes, the waiting dynamics can still be approximated with a single effective rate $\lambda(y)$ (Fig. 2a solid lines; for further details see Supplementary Section 2). Figure 2b shows that these rates increase monotonically with increasing distance from the wall. Notice that this curve does not mirror the algal accumulation at the boundary (Supplementary Fig. 1), a reflection of the fact that proximity to the wall curtails the range of possible swimming directions that algae can have when interacting with the colloid.

Next, we look at the modulation in jump orientation and magnitude. Figure 2c shows the distribution of jump directions $P(\theta, y)$, with motion towards or away from the boundary corresponding to negative and positive values of $\theta$ respectively. Approaching the wall, the distribution develops a marked peak at $\theta = 0$ indicating a strongly anisotropic active motion preferentially parallel to the boundary. This feature reflects the anisotropy in algal dynamics that results from the interaction with the wall[56] and that can be measured up to ~$100\,\mu m$ from the boundary because of the persistence in cells' trajectories[59]. The distribution functions of jump magnitudes are similar to those already reported for micron-sized particles[47,62], with an exponential decay above ~$4\,\mu m$ (Supplementary Fig. 4a). These can be used to calculate the average jump length $\langle l(y) \rangle$ (Supplementary Fig. 4b) which decreases by ~30% from the bulk level within

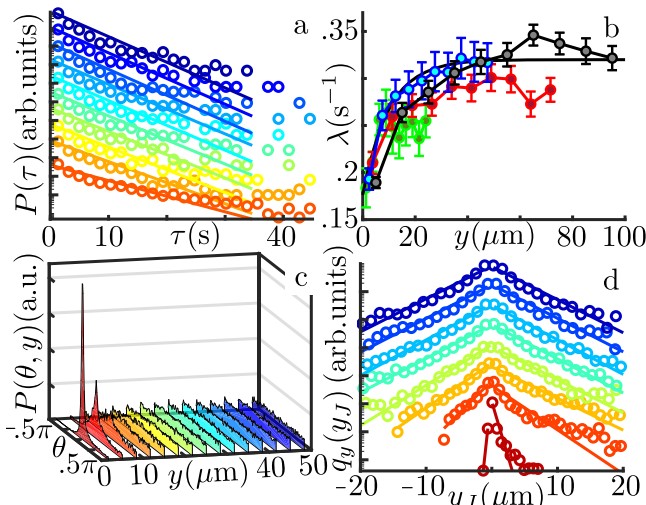

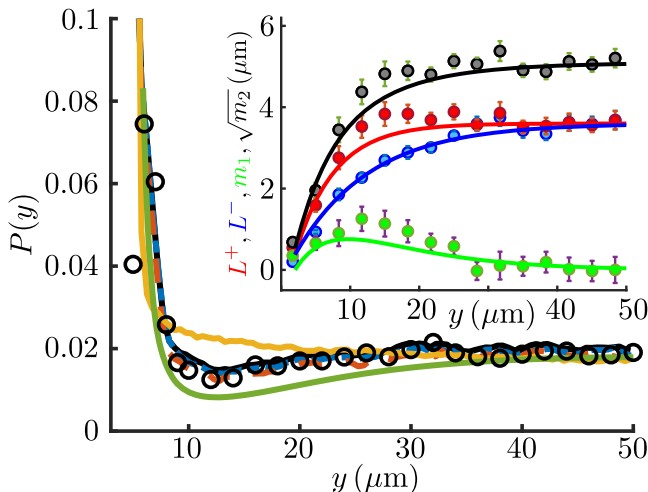

**Fig. 2 | Colloidal jump dynamics. a** Probability distribution function of waiting times between consecutive jumps ($W = 50\,\mu m$, semilog plot) Circles: experiments. Solid lines: exponential fits. The distributions are shifted vertically for clarity. Colours code for position across the channel (red: boundary; blue: channel centre). **b** Characteristic encounter rates $\lambda(y)$ for the four values of $W$, as a function of distance from the channel boundaries. For better comparison, the values have been rescaled to the cell concentration of the $100\,\mu m$-wide channel (see Supplementary Section 3). Circles: experiments. Solid Lines: exponential fits (see Methods). Error bars from the fit uncertainties. **c** Probability distribution function $P(\theta, y)$ for the direction $\theta$ of active jumps vs. distance $y$ from the channel boundary ($W = 50\,\mu m$). **d** Semilog plot of active jump distributions, $q_y(y_J)$, for different distances $y$ from the boundary (Colours as in C). Circles: experiments. Solid lines: exponential fits. The distributions are shifted vertically for clarity.

**Fig. 3 | Comparison between experimental results and the jump-diffusion model.** Steady-state colloidal probability distribution functions in a $2W = 100\,\mu m$ channel: experiments (circles); full dynamics (black solid line); constant encounter rate ($\lambda(y) :=$ bulk value) (teal dashed line); no thermal diffusion ($D_0 = 0$) (purple dashed line); homogeneous and isotropic jump size distribution ($q_y(y_J):=$bulk distribution) (orange solid line); analytical model (olive solid line). The analytical model curve has been shifted downward for clarity. Inset: characteristic jump size away from the boundary ($L_+(y)$, red circles) and toward the boundary ($L_-(y)$, blue circles) together with the first and second moments $m_1(y)$ and $\sqrt{m_2(y)}$ derived from them (Eq. (5)). Solid lines are fitted heuristic analytical functions used to reconstruct the colloidal distribution (see Methods). Error bars propagated from fit uncertainties.

the first $10$–$15\,\mu m$ from the boundary (first $5$–$10\,\mu m$ accessible to the beads). A decrease is indeed expected due to the obvious limitations to the active movement of the colloids imposed by a nearby boundary.

As we are interested in the passive particles' dynamics across the channels, it is useful to reduce the full two-dimensional jump distribution functions to their projection $q_y(y_J)$ along the $y$ axis. Here $y_J$ is the $y$-coordinate of the active displacement and the subscript indicates the distance from the nearest wall (see Fig. 1a for the frame of reference). Figure 2d shows that these distributions are generally well approximated by the combination of two exponentials, one each for positive and negative directions (respectively away from and towards the boundary). The exception is for the positive tails of the distributions closest to the wall, which decay slower than expected from the exponential fit. They will not be considered in the following. The distributions $q_y(y_J)$ can then be approximated as

$$q_y(y_J) = \frac{1}{L_+(y) + L_-(y)} e^{-|y_J|/L_\pm(y)}, \quad 0 \lessgtr y_J, \tag{1}$$

where $L_\pm(y)$ are the characteristic lengths of the exponential fits to positive and negative jumps respectively (see Methods for details on how the characteristic lengths were extracted). As shown in Fig. 3-inset (red and blue symbols and solid lines) these are identical in the core of the channel ($L_+ = L_- \simeq 4\,\mu m$) and decrease to the same small value close to the wall ($\sim 0.5\,\mu m$) as a consequence of the increasing polarisation of the active displacements along the boundary (Fig. 2c). However, the length scale for the transition between boundary and core values is shorter for $L_+(y)$ than for $L_-(y)$ ($\sim 12\,\mu m$ vs. $\sim 25\,\mu m$, see Fig. 3-inset). This behaviour, common to all values of $W$ explored (Supplementary Fig. 5), gives rise to a net drift $L_+(y) - L_-(y) > 0$ towards the bulk of the system (green symbols and solid line, Fig. 3-inset) which, as discussed below, is responsible for the depleted region observed in the experimental

colloidal distributions. The difference between the $L$'s is likely due to a combination of local anisotropy in the distribution of swimming directions of the algae, and the stopping of colloids' active displacements by the wall.

The focus on the active jumps displayed by the passive particles can only be justified if this part of the dynamics is indeed sufficient to capture the experimental steady-state distributions. This was first tested within a 1D numerical simulation of a weakly Brownian particle (diffusivity $D_0$), moving in $y \in [0, 2W]$ and subject to a space-dependent Poisson noise of rate $\lambda(y)$ and value drawn from $q_y(y_J)$ (see Methods Sec. 4.5 for details on the integration scheme). Figure 3 shows that the resulting steady-state distribution (black solid curve) is in excellent agreement with the experimental one (black circles) ($2W = 100\,\mu m$). Numerical simulations also provide a convenient way to explore which elements of the effective colloidal dynamics play the most prominent role. For example, fixing $\lambda(y)$ to the bulk value everywhere leads to a very minimal change in the spatial distribution (Fig. 3, teal dashed line), showing that the spatial dependence of the jump frequency close to the wall is not a major factor in the present case. Similarly, removing completely the background diffusion by setting $D_0 = 0$ leaves the distribution unchanged (Fig. 3, purple dashed line) consistent with the fact that, in our experimental system, colloidal transport is dominated by the Poissonian jumps. On the other hand, simulations that remove the spatial dependence of the jump distributions $q_y(y_J)$ and use their isotropic bulk value everywhere, show a boundary accumulation of colloids that is narrower and higher than the experimental curve and has no intermediate depletion (Fig. 3, orange solid line). This confirms that space-dependent jump anisotropy plays a key role in determining the experimental colloidal distribution.

## Minimal continuum model for the dynamics of the colloids
The numerical validation of the jump-diffusion dynamics motivates a simple analytical model for the evolution of $P_t(y)$, the probability

density of finding a colloid at position $y$ at time $t$:

$$\frac{\partial P_t(y)}{\partial t} = D_0 \frac{\partial^2 P_t(y)}{\partial y^2} - \lambda(y)P_t(y)$$
$$+ \int_{-\infty}^{+\infty} \lambda(y-y_J)P_t(y-y_J)q_{y-y_J}(y_J)dy_J. \qquad (2)$$

In this master equation, local changes in $P_t(y)$ are due either to diffusion (first term in the r.h.s) or to the balance between active jumps from the current position or towards it from elsewhere (second and third terms in the r.h.s., respectively). This continuum equation can also be derived more formally from a stochastic description of the dynamics of single colloids, following Denisov and Bystrik[65] (see Supplementary Section 4). It is worth noting also that Eq. (2) assumes infinitely fast jumps ("teleportation"). While this might seem like a drastic approximation, it is not expected to impact the resulting spatial distributions as long as the typical jump duration is sufficiently shorter than the average waiting time between jumps. This is indeed the case in the current system (~1.2 s vs. ≳3.5 s respectively; see Fig. 2b and Supplementary Fig. 6).

The non-local term in Eq. (2) means that the model is non-tractable for generic kernels $\lambda(y)q_y(y_J)$. In order to make progress, we, therefore, perform a Kramers-Moyal expansion[66] and truncate the series at second-order, providing an approximation of the system at the drift-diffusion level. This type of approximation has already been used successfully to capture the essential features of run-and-tumble bacteria[67], active Brownian particles[68] and active Ornstein-Uhlenbeck particles[69]. This approximation is expected to hold if the characteristic jump size is small compared to the length scale of the heterogeneities in the dynamics. In our case we have $\langle l_{bulk} \rangle \approx 5\,\mu m$ (Supplementary Figure 4b) while a length scale for the heterogeneity can be estimated for example from the $L_\pm$ curves (≳12 μm). As detailed in the Supplementary Materials (Supplementary Section 5) the second-order expansion of Eq. (2) gives a drift-diffusion equation which depends on $\lambda(y)$ and the first and second moments $m_{1,2}(y)$ of $q_y(y_J)$:

$$\frac{\partial P_t(y)}{\partial t} = \frac{\partial}{\partial y}\left[D_{eff}(y)\frac{\partial P_t}{\partial y} - V_{eff}(y)P_t(y)\right]$$
$$D_{eff}(y) = D_0 + \frac{\lambda(y)m_2(y)}{2}$$
$$V_{eff}(y) = \lambda(y)m_1(y) - \frac{1}{2}\frac{\partial}{\partial y}[\lambda(y)m_2(y)] \qquad (3)$$
$$m_n(y) = \int_{-\infty}^{+\infty} y_J^n q_y(y_J)dy_J.$$

Notice that $D_{eff}$ is similar to the asymptotic diffusivity obtained in[47] for homogeneous and isotropic entrainment-dominated transport, with a contribution from the jump events given by the product of jump frequency and variance of jump size. Enforcing no-flux at the boundaries one obtains a closed form for the steady-state solution:

$$P(y) = \frac{B}{D_{eff}(y)}\exp\left(\int^y \frac{\lambda(y')m_1(y')}{D_{eff}(y')}dy'\right), \qquad (4)$$

where $B$ is the normalisation constant. Without a net asymmetry in the dynamics (i.e., when $m_1(y) \equiv 0$), the solution is that of a state-dependent diffusive process with Itô convention for multiplicative noise integration. In our case, however, the first moment of $q_y$ takes significant values in the first ~30 μm from the boundaries (green solid line Fig. 3-Inset) and should not be neglected.

Comparison between Eq. (4) and the experiments is facilitated by having analytical expressions for $\lambda$, $m_1$ and $m_2$. Figure 2b shows that $\lambda$ can be described heuristically by a simple exponential relaxation from the wall to the bulk values (black solid line). As for $m_1$ and $m_2$, the

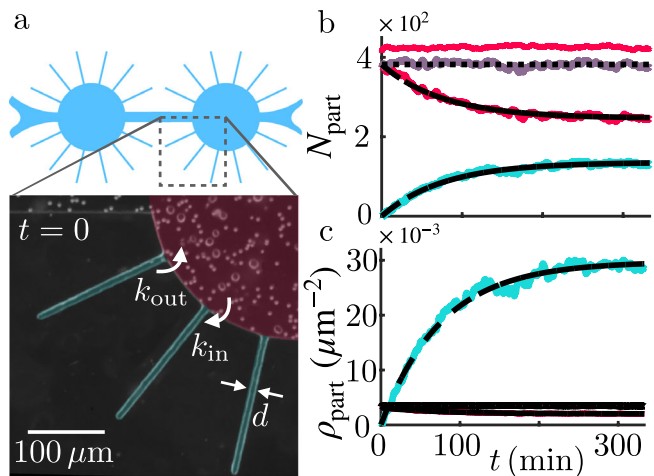

**Fig. 4 | De-mixing of a binary mixture. a** Schematics and details of the microfluidic chamber: 200 μm-radius circular chambers decorated with side channels of width $d = 7$–8 μm. Both cells ($2R = 8$–10 μm) and polystyrene beads ($2a = 6$ μm) are visible in the picture. **b, c** Time evolution of the number of particles and surface number density within the circular chamber ($N_c$, $\rho_c$ red solid line) and the side channels ($N_s$, $\rho_s$ cyan solid line) in presence of the microalgae. Within this time the total number of particles in the chamber remains constant (purple solid line). Dashed/dash-dot lines are a fit to the first order kinetics (see Supplementary Section 6). Dotted lines indicate the initial values of $N_c$ and $\rho_c$. Dashed red line corresponds to the control experiment without microalgae.

description of $q_y(y_J)$ given by Eq. (1) implies that

$$\begin{aligned} m_1(y) &= L_+(y) - L_-(y) \\ m_2(y) &= 2\frac{L_+^3(y)+L_-^3(y)}{L_+(y)+L_-(y)}. \end{aligned} \qquad (5)$$

Given that $L_\pm(y)$ themselves are well approximated by exponentials, this affords analytical approximations also for $m_{1,2}(y)$ (Fig. 2c-Inset). Armed with this description we can now compare the experimental curves with the prediction from the approximate jump-diffusion model. The olive-coloured curve in Fig. 3 shows that the theoretical distributions recapitulate extremely well the experimental ones. On one hand, this provides a justification *a posteriori* for the modelling approach taken above. On the other it reveals that, at a continuum level, the dynamics of colloids in an active bath can be reduced to two quantities: the first and the second moments of the active displacements. This provides an intuitive way to conceptualise the complex dynamics of colloidal particles within an active suspension.

## Induced de-mixing of the passive particles

What we learned so far on the colloids' emergent active dynamics can be harnessed to induce the de-mixing of our active-passive suspensions, as a first step towards more complex control of microscale cargo transport. The idea is to include a confining boundary which acts as a kind of selective membrane, letting the beads cross but not the algae. This is achieved by considering circular chambers decorated with 190 μm-long side channels whose width $d$ (Fig. 4a) is smaller than the swimmer diameter but larger than the bead diameter (here taken as $2a = 6\,\mu m < d = 7$–8 μm $< 2R = 8$–10 μm). Due to the active dynamics of the colloids, we expect them to be pushed inside the side channels by the algae, therefore depleting the particles in the main chamber and spatially separating the active and passive species. As shown Fig. 4b (see also Supplementary Movie 4) this is exactly what we obtain, with the number of particles in the circular chamber $N_c$ (red solid line) quickly decreasing and the number in the side channels $N_s$ (cyan solid line) increasing. Notice that the total number of particles $N_t = N_c + N_s$ in the system (purple solid line) remains constant over the duration of the experiment. The process follows a first order kinetic with rates $k_{in}$

and $k_{out}$ for transitions towards or out-of the side channels respectively (see Supplementary Section 6). The occupancy numbers follow an exponential relaxation with time-scale $1/(k_{in} + k_{out}) = 77 \pm 4$ min and stationary values $N_c/N_t = k_{out}/(k_{in} + k_{out}) = 0.64 \pm 0.02$ and $N_s/N_t = k_{in}/(k_{in} + k_{out}) = 0.36 \pm 0.02$, leading to estimates for the two rates of $k_{in} = (7.8 \pm 0.6) \times 10^{-5}\,s^{-1}$ and $k_{out} = (13.9 \pm 0.8) \times 10^{-5}\,s^{-1}$. The latter results from the passive dynamics of the colloids within the side channels, which depends on their thermal diffusivity, potential electrostatic interactions with the boundaries, short-range hydrodynamic interactions between the beads[70], channel design, etc[71]. The former, instead, is the consequence of active colloidal displacements. Assuming no colloid/wall attraction within the side channels (as confirmed by our control experiment, see below), it encapsulates the intrinsic out-of-equilibrium nature of the system, a property which is immediately clear from the significant difference in steady-state densities between the side channels ($\rho_s = 0.0295 \pm 0.007$ beads/μm²) and in the main chamber ($\rho_c = 0.0019 \pm 0.0001$ beads/μm²) (Fig. 4c). The value of $k_{in}$ can be compared with an estimate derived from the effective colloidal dynamics from Eq. (3) (see Supplementary Section 7). The mean first passage time of a colloid to the entrance of any of the side channels returns a rate of $k_{in}^{V,D} = (7.408 \pm 0.1) \times 10^{-5}\,s^{-1}$ which agrees very well with the experimental value. This shows that the simplified advection-diffusion model for the passive particles is a good description not just for time-independent properties like the steady-state distribution of Fig. 3, but also for dynamic ones like the rate of active cargo transport to the side channels. It is instructive also to use the model to explore the contributions of the different parts of the dynamics. Hence we see that setting $V_{eff} \equiv 0$ and $D_{eff} \equiv 3.55$ μm² s⁻¹, the effective diffusivity far from the boundaries, one obtains a rate $k_{in}^D = (16.2 \pm 4.1) \times 10^{-5}\,s^{-1}$. The stark difference between $k_{in}^{V,D}$ and $k_{in}^D$ suggests that, in our case, it is not sufficient to know the behaviour of the active-passive suspension in the bulk of the chamber. The behaviour close to the boundary, where the inhomogeneities are found, is essential to grasp the dynamics of the de-mixing process. Repeating the same experiment only with colloids shows indeed no appreciable filling of the side channels within the duration of the experiment (~10 h; see Supplementary Movie 5) and the particle density in the main chamber remaining constant (Fig. 4b red dashed line).

Although the microfluidic chip in Fig. 4 leads to a much higher concentration of colloids in the side channels than in the main chamber, this still involves only ~36% of the colloids. A different design, however, could maximise the fraction of de-mixed colloids instead of their concentration. For example, modifying the current design to have two identical chambers connected by a narrow gap of size $d = 7-8$ μm should produce a fraction of de-mixed colloids given by $k_{in}/\left(k_{in} + k_{in}^{th}\right) \simeq 95\%$. The design could then be tailored to different needs.

## Discussion

In this study we used an active-passive system as a minimal model to investigate the active transport of passive microscopic objects. Leveraging the well-known boundary accumulation of microswimmers, we saw that the emerging interactions between active and passive components are also heterogeneous and anisotropic. In turn, they lead to a complex spatial distribution of colloids and to the possibility of de-mixing the suspension. These properties are well described by a simple advection-diffusion model for the passive species, based only on the first and second moments of their ensuing active displacements. This type of approximation, then, provides a degree of universality in the description of the dynamics of active systems including purely active and active-passive species. From the targeted delivery of chemicals, biomarkers or contrast agents at specific locations in the body to the remediation of water and soils, artificial and biological microscopic self-propelled particles hold a great potential to address critical issues in areas ranging from personalised medical care to environmental

sustainability[72,73]. Development in these areas will depend on our ability to use and control micro/nano-swimmers to perform essential basic functions, such as sensing, collecting and delivering passive cargoes in an autonomous, targeted and selective way[74]. Within this perspective, we have shown here that heterogeneous activity fields can be employed to control the fate of colloidal objects in a manner similar to biological systems, where the spatial modulation of the activity can control, for example, the positioning of organelles within cells[14–16]. Although this was achieved here through simple confinement, many microswimmers possess -or can be designed to have- the ability to respond to a range of external physico-chemical cues or stimuli[51–54]. We expect that similar phenomenologies should also emerge whenever the active particles in a mixed system autonomously generate a space-dependent concentration profile. Our work paves the way to using these to dynamically manipulate the fate of colloidal cargoes by externally altering microswimmers' dynamics. Advance in this area will require to understand how to predict the active displacements of passive cargoes directly from the dynamics of microswimmers[75], a theoretical development that we leave for a future study.

## Methods

### Microorganism culturing

Cultures of CR strain CC-125 were grown axenically in a Tris-Acetate-Phosphate medium[76] at 21 °C under periodic fluorescent illumination (16h light/8h dark cycles, 100 μEm⁻² s⁻¹, OSRAM Fluora). Cells were harvested at ~5 × 10⁶ cells/ml in the exponentially growing phase, then centrifuged at 10 × g for 10 min and the supernatants replaced by DI-water containing the desired concentration of polystyrene colloids (Polybead® Carboxylate Microspheres, radius $a = (5 \pm 0.5)$ μm, Polysciences).

### Microfluidics and microscopy

The mixed colloid-swimmer solution was injected into polydimethylsiloxane-based microfluidic channels, previously passivated with a 0.5% w/w Pluronic-127 solution to prevent sticking of the particles to the surfaces of the chips. The microfluidic devices were then sealed with Vaseline to prevent evaporation. The channels of varying widths were observed under bright-field illumination on a Zeiss AxioVert 200M inverted microscope. A long-pass filter (cutoff wavelength 765 nm) was added to the optical path to prevent phototactic responses of the cells. For each channel width stacks of 80,000 images at ×10 magnification were acquired at 10 fps with an IDS UI-3370CP-NIR-GL camera.

### Data analysis: trajectories

Particle Trajectories were digitised using a standard Matlab particle tracking algorithm (The code can be downloaded at http://people.umass.edu/kilfoil/downloads.html). After the particle trajectories were reconstructed, the large jumps were isolated using a combination of move size and directional correlation along the trajectory (see detailed protocol in[47,63]).

### Data analysis: fitting of $L_+(y)$, $L_-(y)$ and $\lambda(y)$

The experimental curves $L_+(y)$, $L_-(y)$ and $\lambda(y)$ have been fitted by the following analytical functions:

$$L_+(y) = (L_+^w - L^b)e^{-\frac{y-y_{col}}{l^s}} + L^b, \tag{6}$$

$$L_-(y) = (L_-^w - L^b)e^{-\frac{y-y_{col}}{l^s}} + L^b, \tag{7}$$

$$\lambda(y) = (\lambda^w - \lambda^b)e^{-\frac{y-y_{col}}{l^\lambda}} + \lambda^b, \tag{8}$$

**Table 1 | Fit parameters (µm) for Eqs. (6), (7) and (8) and their standard error**

| $L^b$ | $l^+$ | $l^-$ | $\lambda^w$ | $\lambda^b$ | $l^\lambda$ |
|------|------|------|------|------|------|
| 3.70 | 5.05 | 10.8 | 0.237 | 0.315 | 8.20 |
| 0.079 | 0.764 | 0.978 | 0.0046 | 0.0040 | 1.278 |

**Table 2 | Limits in $y_J$ for the $y$-dependent distributions of jumps perpendicular to the channel boundaries**

| $y$(µm) (bin centre) | 5.0 | 8.3 | 11.7 | 15.0 | 18.4 | 21.7 | 25.0 |
|------|------|------|------|------|------|------|------|
| lim($y$) (µm) | 7 | 12 | 12 | 15 | 15 | 18 | 20 |
| $y$(µm) (bin centre) | 28.4 | 31.7 | 35.0 | 38.4 | 41.7 | 45.0 | 48.4 |
| lim($y$) (µm) | 20 | 20 | 20 | 20 | 20 | 20 | 20 |

with $L_+^w = L_-^w = 0.375\,\mu m$ fixed. The best fits (dotted lines in Figs. 2 and 3) are given by the parameters in Table 1.

The experimental curves $L_+(y)$ and $L_-(y)$ were obtained from the experimental jump size distributions $q_y(y_J)$ by fitting the logarithm of each side (i.e., positive and negative $y_J$) with affine functions. Because these distributions deviate from simple exponentials as we get closer to the boundary (Fig. 2d), the fits were restricted to $|y_J| < \text{lim}(y)$ in order to extract the characteristic length scales of the initial decay of the distributions. The resulting fits are shown in Fig. 2d (solid lines). The limits for the fits of the experimental distributions were fixed as in Table 2.

### Numerical simulations

One hundred trajectories composed of 10,000 time-steps (with $\delta t = 0.1\,s$, equal to the acquisition period of the experiments) were simulated for each condition. At each time step an acceptance-rejection scheme is performed to decide whether the particle undergoes a jump: (i) a (uniform) random number is first drawn from the interval [0,1], (ii) if this number sits within the interval $[(1-\lambda(y)\delta t)/2; (1+\lambda(y)\delta t)/2]$, the particle undergoes a jump whose size is taken from the experimental distribution $q_y(y_J)$ (using inverse transform sampling), (iii) otherwise the particle simply undergoes Brownian motion with diffusivity $D_0 = 0.0439\,\mu m^2\,s^{-1}$ (the theoretical bulk diffusivity for a 5 µm-radius particle at room temperature). Upon reaching a boundary jump moves undergo a stopping boundary condition, stopping at ±5 µm (the particle's radius) from the boundary (this is to be as faithful with our experimental observations as possible). For simplicity, diffusive moves that cross the boundary undergo a reflective boundary condition. More accurate boundary implementations for the diffusive motion do not lead to different results as the impact of thermal diffusivity in the channel experiments is negligible.

## Data availability

The main data used in this study are available in the Zenodo database with https://doi.org/10.5281/zenodo.6866462.

## Code availability

The codes used in this study are available in the Zenodo database with https://doi.org/10.5281/zenodo.6866462.

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

## Acknowledgements

R.J. would like to thank Denis Bartolo, Jean-François Rupprecht and Jérémie Palacci for enlightening discussions. This work was

supported in part through the Ramón y Cajal Programme (RYC-2018-02534; MP), the Spanish Ministry of Science and Innovation (Grant PID2019-104232GB-I00 funded by MCIN/AEI/ 10.13039/501100011033; M.P. and I.T.), the Margalida Comas Programme (PD/007/2016; R.J.), the Juan de la Cierva Incorporación Programme (IJCI-2017-32952; R.J.), the Junior Research Chair Programme (ANR-10-LABX-0010/ANR-10-IDEX-0001-02 PSL; R.J.).

## Author contributions

R.J., I.T. and M.P. conceived and designed the experiments. R.J. and S.W. performed the experiments. R.J., S.W. and M.P. analysed the data. R.J. developed the analytical model and S.W. performed the numerical simulations. All authors discussed the results and wrote the article.

## Competing interests

The authors declare no competing interests.
