## [Peer Review File · Nature Communications]

Confinement-induced accumulation and de-mixing of microscopic active-passive mixturesThis manuscript has been previously reviewed at another journal that is not operating a transparent peer review scheme. This document only contains reviewer comments and rebuttal letters for versions considered at *Nature Communications*.

REVIEWERS' COMMENTS

Reviewer #1 (Remarks to the Author):

I have read the updated manuscript submitted by the authors as well as their answers to the three referee reports. I think the manuscript has been clarified and the questions raised by the referees properly addressed. The authors demonstrate experimentally how a confined active bath will act as a non-uniform source of active noise for passive tracers, hence allowing the control of their steady-state distribution. I think this is an important result, whose connection to the existing literature is now clearly done. The universality of the mechanism is convincing and I do not think that understanding the microscopic details of the interactions between algae and colloids and walls---as asked by referee 2---is that crucial. Much like the detail of the exchange interaction is unimportant when it comes to understanding the phase transition towards ferromagnetism, the authors work capture the emergent physics of their system, which is the most salient aspect of their experiments.

Overall, I think this is a very nice work that will have a large impact and will appeal to a wide audience. I thus recommend its publication in Nature Communication.

Reviewer #4 (Remarks to the Author):

The work deals with the interaction between active and passive particles in a confined environment. The authors find that in the presence of active particles, the passive particles organise with a non-Boltzmann probability distribution due to the continuous interaction with the active particles. They also characterise and describe with a phenomenological mathematical model the motion of the passive particles. They finally employ these principles to push the passive particle into a microscopic sieve.

I find the overall work interesting and driven by a very good novel idea — to control the distribution of passive colloidal particle by activity gradients. I'm therefore inclined to see this published in Nature Communication (also in light of the comments and replies to the other three referees).

If I can add some suggestions, which should be at least acknowledged in the discussion:

1. The last example of the sorting raises some concerns because the active particles cannot really get into the side channels because of their size. It'd been more convincing if this were not the case.

2. The description of the model with multiplicative noise could be grounded on a more solid foundation, taking into account that it is a macroscopic model, whose underlying details are homogenised. In this sense, it's very important to control for the assumptions that can be made in an active system.

Minor comment:

- line 95: what are "weakly Brownian" colloids?

Reviewer #1 (Remarks to the Author):

I have read the updated manuscript submitted by the authors as well as their answers to the three referee reports. I think the manuscript has been clarified and the questions raised by the referees properly addressed. The authors demonstrate experimentally how a confined active bath will act as a non-uniform source of active noise for passive tracers, hence allowing the control of their steady-state distribution. I think this is an important result, whose connection to the existing literature is now clearly done. The universality of the mechanism is convincing and I do not think that understanding the microscopic details of the interactions between algae and colloids and walls---as asked by referee 2---is that crucial. Much like the detail of the exchange interaction is unimportant when it comes to understanding the phase transition towards ferromagnetism, the authors work capture the emergent physics of their system, which is the most salient aspect of their experiments.

Overall, I think this is a very nice work that will have a large impact and will appeal to a wide audience. I thus recommend its publication in Nature Communication.

We thank reviewer #1 for supporting publication of our manuscript in Nature Communications.

Reviewer #4 (Remarks to the Author):

The work deals with the interaction between active and passive particles in a confined environment. The authors find that in the presence of active particles, the passive particles organise with a non-Boltzmann probability distribution due to the continuous interaction with the active particles. They also characterise and describe with a phenomenological mathematical model the motion of the passive particles. They finally employ these principles to push the passive particle into a microscopic sieve.

I find the overall work interesting and driven by a very good novel idea — to control the distribution of passive colloidal particle by activity gradients. I'm therefore inclined to see this published in Nature Communication (also in light of the comments and replies to the other three referees).

We thank reviewer #4 for their positive assessment of our work and for supporting publication in Nature Communications. We reply to the comments made by the reviewer below.

If I can add some suggestions, which should be at least acknowledged in the discussion:

1. The last example of the sorting raises some concerns because the active particles cannot really get into the side channels because of their size. It'd been more convincing if this were not the case.

As replied to point B of reviewer 3, the point of having side channels that can accommodate the colloids but not the algae is to create two regions with different concentrations of active particles in order to create a net flux of the passive components from the region of high activity to the region of low activity (zero activity, in this case). Of course, the inhomogeneity in active species concentration does not emerge spontaneously but is imposed externally thanks to the small width of the side channels. The central point here is not how this inhomogeneous concentration is created, but rather the effect that it has on the passive component. The accumulation of the colloids in the side channels is caused precisely by the spatial variation in the concentration of active particles (and therefore in the activity intrinsic within the system). Without this spatial variation in activity, we would go back to a case that has already been studied since 2001 of a homogeneous and isotropic active bath. In that case there is no accumulation of colloids.

The way we use here the size difference between colloids and microorganisms is, in a sense, accidental. It is only a means to have an external field that induces a spatial variation in the properties of the active species. Of course this could have been implemented in a different way for example with a localised environmental stimulus (light, chemoattractants, etc) triggering a microbial response (taxis), but we opted instead for something that in our opinion is the simplest and most robust choice.

We feel like this point is already being acknowledged in the discussion of the manuscript where we wrote: "Although this was achieved here through simple confinement, many microswimmers possess - or can be designed to have- the ability to respond to a range of external physico-chemical cues or stimuli [51-54]. We expect that similar phenomenologies should also emerge whenever the active particles in a mixed system autonomously generate a space-dependent concentration profile. »

2. The description of the model with multiplicative noise could be grounded on a more solid foundation, taking into account that it is a macroscopic model, whose underlying details are homogenised. In this sense, it's very important to control for the assumptions that can be made in an active system.

We hope we understand the point made by the referee and apologise in advance if this is not the case and we have misunderstood the comment.

In the manuscript we use two (connected) models. The first is the macroscopic continuum model that we see in Eq.(2) and that we proceed to expand following to the usual Kramers-Moyal expansion. The second is a single-particle model given by Eq.S21 in the Supplementary Material. The former is a deterministic partial differential equation, so there is no multiplicative noise to be further defined. We therefore assume that the referee is talking about Eq.S21.

At the point in the Supplementary Material when we introduce Eq.S21, it is well established that the effective dynamics of the colloids can be described by a drift diffusion model at least insofar as the steady state equilibrium distribution of colloids goes. We are interested here in take this model at face value to see whether a *dynamic* quantity like the rate of escape (k_{in}) can also be predicted by the same model. Of course we cannot expect a drift-diffusion model to be a good description of the dynamics at short timescale. For one, it obviously does *not* reproduce the colloids' jumps. The question is whether the model is good for dynamic observables that depend on timescales long compared to the waiting time between successive jumps (timescales that cover several jumps). This is similar to describing a run and tumble bacterium as a diffusing particle with an effective diffusivity. The simplified model captures the behaviour of the bacterium only at times long compared to the run time. Along the same line, we can compare ideal polymer models given by a freely jointed chain and by a Gaussian chain. They are not the same at length scales comparable to the bond length, but they behave the same at longer length scales. We have now amended the text on page S5 of the supplementary material to clarify this point (added text in orange):

"[...] In order to include them in the estimate of the escape, we perform a numerical simulation of a colloid subject to the space-dependent effective diffusivity and drift used in the KM model. *We expect this to be a minimal model for the dynamics of individual colloids, for timescales long compared to the waiting time between successive jumps. To simulate the discrete dynamics, we adapt to our case the so-called Milstein method:[...]*"

Notice that Equations S21 are discrete time equations, not differential equations, and therefore are well defined. They correspond to the so-called Milstein method, a well known numerical technique for the approximation of drift-diffusion stochastic differential equations with multiplicative noise. We now explicitly mention it in the Supplementary Material as well, see amendment above. The interpretation of the multiplicative noise depends on the parameter α that determines the drift in Eq.S21, as described explicitly with great clarity in the reference that we cite on page S5 (reference [8] of the Supplementary Material). Our results show that the standard Langevin drift-diffusion equation that one can write down appears to be a good description of the dynamics for both common choices of α (Itô and Stratonovich), although the Stratonovich interpretation gives a numerical value that is closer to the experimental one. The reason why this might be the case is likely to be found from a formal derivation of the drift-diffusion Langevin description from a more detailed description of the actual dynamics, e.g. a jump-diffusion model similar to the one we discuss in Eq.S1 (from Denisov and Bystrik, reference [4]). We believe that this is interesting but out of the scope of the current paper.

Minor comment:

- line 95: what are "weakly Brownian" colloids?

"Weakly Brownian" is a term that is commonly used to describe colloidal particles that are of a size range such that their thermal motion provides a noise that is small but not entirely negligible compared to other phenomena that are being studied. Within colloidal science, weakly Brownian is usually used to indicate colloids of a size around 10 μ m. For these, their Brownian time, i.e. the time required to diffuse a distance equal to their size, is just observable within the usual timescale of experiments.